# An Exploration of the Design Space of Speech-Conditioned Large Language Models (SLMs)

## Abstract

Efforts to enable Large Language Models (LLMs) to understand human speech and following spoken instructions have led to the development of several Speech-Conditioned Large Language Models (SLMs). While these models have demonstrated success on various speech-related tasks, such as automatic speech recognition (ASR), speech-to-text translation (ST), the design space of SLMs has not been thoroughly explored. With the goal to build an SLM that is able to follow spoken instructions (to understand non-speech audio is out-of-scope for this work), we revisit key design choices for SLMs in this work. Our experiments allow us to gain insights into how these choices impact the ability of SLMs to understand human speech and follow spoken instruction and how we could optimize important design choices to achieve better results. Surprisingly, our experiments reveal that existing SLMs struggle to follow spoken instructions or respond to speech inputs, even for simple queries like "`who has been to the moon?`". Our findings indicate that using spoken instruction following data is crucial for improving such a capability. Leveraging this insight, we build a synthetic spoken instruction following dataset and train our SLM with it. Combining the findings from our other experiments, we provide an effective recipe for developing SLMs. Our model, called SiM, not only achieves strong ASR performance but also significantly outperforms existing SLMs in spoken instruction following.

## 1 Introduction

Since the emergence of Large Language Models (LLMs) (Jiang et al., 2023; Team et al., 2023; Brown, 2020; Radford et al., 2019; Touvron et al., 2023a;b; Javaheripi et al., 2023; Zhou et al., 2024; Li et al., 2023b; Tunstall et al., 2023), researchers have been striving to endow them with the ability to understand and follow multi-modal instructions, which may contain images, videos, speech, and non-speech sounds, in addition to texts (Li et al., 2024a;b;c; Lin et al., 2024). As speech is one of the most frequently used modes of communication among humans, developing models capable of understanding speech and respond to spoken queries, *i.e.*, follow spoken instructions, is of great importance. Such a capability is indispensable for building models with natural human-computer interaction interfaces, enabling a wide range of applications such as voice assistants, speech-to-text transcription, and language learning tools. Speech language models have the potential to revolutionize how we interact with machines, making it more intuitive and accessible for users of all backgrounds. Furthermore, these models can facilitate communication for individuals with disabilities, bridging the gap between humans and technology.

An SLM takes one or more speech clips (and optionally texts) as inputs and generates text responses accordingly (Tang et al., 2023; Chu et al., 2023; 2024). Witnessing the great success of LLMs and vision-conditioned LLMs (VLMs) (Liu et al., 2024b;a; Xue et al., 2024; Fang et al., 2024; Wang et al., 2024c;b; Chen et al., 2024), which adopt simple Transformer-based architectures and next-token prediction loss, most SLMs have followed a similar approach.

They leverage pre-trained speech encoders, such as Whisper (Radford et al., 2023) or Conformer (Gulati et al., 2020), to extract speech features from the input audio. These features are treated as speech token embeddings and then projected into embedding space of text tokens. The projected features are fed into an LLM alongside text token embeddings. While being straightforward, this approach enables SLMs to be trained using next-token prediction loss, harnessing the capabilities of both pre-trained speech encoders and LLMs. Such an approach leads to the success development of several SLMs, including SALMONN (Tang et al., 2023) and Qwen Audio Chat (Chu et al., 2023).

While SLMs share many similarities, they differ from one another in various aspects, such as model architecture and training data. However, drawing meaningful comparisons between existing SLMs is challenging due to the lack of consistent experimental settings across different works. As a result, it is difficult to determine which design choices are truly superior and how they impact model performance. This gap highlights the need for comprehensive research that thoroughly examines the design space of SLMs and systematically compares their design choices under comparable settings. Motivated by this need, we conduct rigorous experiments with consistent experimental setups to investigate the design space of SLMs, with the goal of identifying important design choices, understanding their influence on model performance, and exploring ways to optimize these choices for better overall performance.

In this work, we begin with the commonly used "speech feature-as-token" approach. We first explore different axes of alignment training, which aims to align features extracted by pre-trained speech encoders with embeddings of text tokens, a process referred to as stage 1 training or pre-training in some literature. We systematically investigate various aspects, including architecture, data, and the choice of LLM, in a step-by-step manner. Subsequently, we delve into how instruction tuning can enhance our models' ability to follow speech instructions. Surprisingly, we found that existing methods lack such an ability due to the absence of spoken instructions in their instruction tuning data. While these models are trained using thousands of hours of speech data, the speech clips serve as context, and the instructions are provided in text form. Based on this finding, we build a synthetic spoken instruction following dataset containing 50K samples with spoken instructions and demonstrate that this significantly improves our model's capability to follow speech instructions. In summary, our key contributions are as follows::

- We conduct rigorous experiments to thoroughly explore the design space of SLMs and compare different design choices.
- Benefiting from our exploration, we provide an effective recipe for training our SLM, SiM, which achieves strong ASR performance and significantly outperforms existing models in spoken instruction following.

## 2 PRELIMINARIES

In this section, we introduce the model architecture, evaluation data and default training settings of our model.

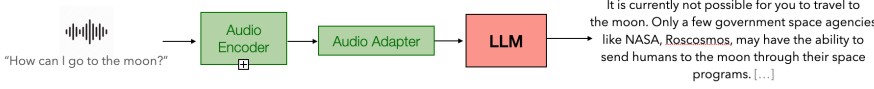

Figure 1: Architecture of our model. Following recent SLMs (Tang et al., 2023; Chu et al., 2023; 2024), we adopt an (encoder, adaptor, LLM) architecture. More details are provided in Section 2.1.

### 2.1 MODEL ARCHITECTURE

Following SALMONN (Tang et al., 2023), Qwen Audio Chat (Chu et al., 2023), and recent VLMs, including LLaVA, Prismatic VLMs, etc., we adopt the encoder-adaptor-LLM architecture. Specifically, this architecture is composed of three components: a speech encoder,

an adaptor and an LLM. The speech encoder encodes speech clips, whose output is then projected into the embedding space of the LLM by the adaptor. Formally, given an audio clip $\mathbf{x}$, a speech encoder $\mathcal{E}$ encodes it into $\mathbf{e} = \mathcal{E}(\mathbf{x})$. Here, $\mathbf{e} \in R^{c' \times t}$. $c'$ and $t$ represent the dimension of speech features and number of speech feature vectors. An adaptor $\mathcal{A}$ projects speech features into the embedding space of text tokens via $\mathbf{a} = \mathcal{A}(e)$, where $\mathbf{a} \in R^{c \times t}$. $c$ denote the dimension of token embeddings of the LLM. The projected features $\mathbf{a}$ are treated as audio token embeddings and are (optionally) concatenated with text token embeddings before being fed into the LLM.

## 2.2 Default Training Settings

We introduce the default training settings with which we start. Most of these settings will be examined through experiments and will be changed if we find a better one.

We use Llama3 8B Instruct, which is a strong open source model, as our LLM. While some existing SLMs are built upon Vicuna[1] or Llama2, results obtained on a more recent and stronger LLM will be more valuable for future researchers. We use a 2-layer Multi-Layer Perceptron (MLP) as the adaptor. We adopt pre-trained Whisper Large v2 (Radford et al., 2023) as our speech encoder (Chen et al., 2022; Baevski et al., 2020; Schneider et al., 2019; Hsu et al., 2021) and process raw speech clips following Whisper.

For alignment training, we start with LibriSpeech dataset and use its train-clean 360 hour subset, denoted as LibriSpeech 360, due to its wide adoption and high quality. We will introduce instruction tuning data we used in later sections as we build our own dataset. We perform full finetuning when optimizing paramters of the LLM.

## 2.3 Evaluation Suite

For alignment training, as we train our models to perform automatic speech recognition (ASR), we evaluate their ASR performance on LibriSpeech dataset (test-clean subset) and adopts word error rate (WER) as the evaluation metric. Following common practice (Radford et al., 2023), we normalize both reference transcripts and model predictions before computing WER.

For instruction tuned models, we evaluate their ASR performance in the aforementioned way. More importantly, we measure how well they follow spoken instructions on Alpaca Audio dataset (Wang et al., 2024a), OpenHermes Audio dataset (Wang et al., 2024a) and LLaMA Questions dataset (Nachmani et al., 2023). These datasets are composed of open-ended questions and reference answers. Model responses to spoken instruction from Alpaca Audio dataset and OpenHermes Audio dataset are judged by Llama3 70B Instruct (Dubey et al., 2024). It assigns a score $s, s \in \mathbb{Z} \cap [0, 5]$ to each model response. A score of 0 indicates that the repsonse is of low quality, while a score of 5 are assigned to high quality responses. More details can be found in (Wang et al., 2024a). Accuracy is used as the evaluation metrics to measure model performance on LLaMA Questions dataset (Nachmani et al., 2023). We refer our readers to (Nachmani et al., 2023) for more details.

## 3 Experiments - Exploration of Design Space of SLMs

In this section, we conduct experiments to explore the design space of SLMs. We start by examining design choices of alignment training and then proceed to examine those of instruction tuning.

### 3.1 Alignment Training

#### #1. Choice of Adaptor Architecture

Various adaptor architectures have been designed for vision-conditioned language models (VLMs), such as Q-Former (Li et al., 2023a), MLP (Liu et al., 2024b), C-abstractor (Cha

---

[1]Vicuna can not be commerially used due to its use of GPT generated data.

| Adaptor Architecture | # Audio Tokens | ATR | WER ↓ |
|---|---|---|---|
| 2-layer MLP | 1,500 | 1× | 4.9% |
| 2-layer ConvNet | 375 | 4× | 5.6% |
| 3-layer ConvNet | 188 | 8× | 7.9% |
| 3-layer Q-Former | 186 | 8× | >20.0% |
| 3-layer Q-Former | 88 | 16× | >80.0% |
| 3-layer Windowed Q-Former | 372 | 4× | 8.0% |
| 3-layer Windowed Q-Former | 186 | 8× | 9.5% |
| 3-layer Windowed Q-Former | 88 | 4× | 10.4% |

Table 1: Comparison between four adaptor architectures: (1) Multi-Layer Perceptron (MLP), (2) Convolutional Network (ConvNet), (3) Q-Former, (4) Windowed Q-Former. For each adaptor, we present its number of *output* audio tokens (# Audio Tokens), audio token reduction rate (ATR) and word error rate (WER). ATR is the ratio between the adaptor's number of input audio tokens and its number of output audio tokens.

et al., 2024), SVA (Tong et al., 2024). Different adaptors have also been proposed or adopted by SLMs, *e.g.*, Windowed Q-Former (Tang et al., 2023). Here, we compare four types of adaptor architectures:

1. MLP: a Multi-Layer Perceptron (MLP)

2. ConvNet: a Convolutional Network (ConvNet) which is mainly composed of 1D convolution layers. We design ConvNet drawing inspiration from C-abstractor (Cha et al., 2024). Please refer to Appendix for detailed architecture.

3. Q-Former: our implementation of Querying Transformer (Q-Former) (Li et al., 2023a)

4. Windowed Q-Former: our implementation of Windowed Q-Former (Tang et al., 2023)

We compare eight different variants of the four adaptor architectures in Table 1. We see that a simple 2-layer MLP achieves the lowest WER. A 2-layer ConvNet achieves slightly higher WER, while having a token reduction rate of 4. In other words, the number of *output* speech tokens is one forth of its *input* speech tokens. This greatly reduces the computational cost, as the computational complexity of self attention layers (Vaswani, 2017) in an LLM is quadratic to the number of input tokens. Models with a Q-Former-based adaptor suffer from training instability. While Windowed Q-Formers performs worse than their ConvNet counterparts with similar token reduction rates.

Conclusion #1: 2-layer MLP adaptor achieves the best performance without token reduction. If token reduction is needed, ConvNet adaptors are good choices.

#2. Choice of Trainable Module(s)

| Trainable Module | # Trainable Parameters | WER ↓ |
|---|---|---|
| Adaptor Only | 0.02B | 4.89% |
| Adaptor + LLM | 8.05B | 4.67% |
| Adaptor + Whisper | 0.66B | >15.0% |

Table 2: Word error rates (WER) of three variants of our model. (1) Adaptor only: Only adaptor are trained. Whisper encoder and LLM are kept frozen. (2) Adaptor + LLM: adaptor and LLM are trained/finetuned. Whisper encoder is kept frozen. (3) Adatpor + Whisper: adaptor and Whisper encoder are trained/finetuned. LLM is kept frozen.

In addition to the adaptor whose parameters needs to be trained from scratch, whether finetuning weights of other composing modules of SLMs leads to better performances is

an open question. This is also a debatable question for VLMs. Two different works draw contracting conclusions. One (Tong et al., 2024) argues for the joint learning of the adaptor and its pre-trained visual encoder during alignment learning, another (Karamcheti et al., 2024) claims that finetuning the visual encoder leads to terrible model performance. In this work, we examine three choices: training the adaptor only, jointly training/finetuning the adaptor and the LLM, jointly training/finetuning the adaptor and the Whisper encoder.

We can see from Table 2 that jointly learning the adaptor and Whisper encoders results in unsatisfactory model performance, *i.e.*, >15% WER. As weights of the adaptor are randomly initialized, finetuning Whisper encoder to work with the adaptor may undermine the speech encoding capabilities of Whisper encoder. Training/finetuning the adaptor and the LLM leads to the lowest WER of 4.67%, 0.22% lower than that achieved by training the adaptor only, which is surprisingly good considering only less that **0.23%** of our model's parameters are being trained. The 0.22% lower WER comes at a cost of about two times longer training time and the potential of harming the LLM's capability to handle text prompts. Hence, we choose to train the adaptor only and leave an in-depth exploration of joint adaptor and LLM training for future work.

# 3. Choice of Masking Strategy

Whisper pads speech clips shorter than 30 seconds to 30 seconds. While Whisper encoder (and also Whisper decoder which is not used in this work) attends to all speech tokens, whether letting the LLM attends to padded speech tokens or not is an open question. We compare two masking strategies. (1) We do not mask any speech tokens (denoted as None). In other words, similar to Whisper decoder, the LLM attends to all speech tokens. (2) We mask padded speech tokens, which will not be attended to by the LLM.

| Masking Strategy | WER ↓ |
|---|---|
| None | 4.89% |
| Padded Speech Tokens | 4.93% |

Table 3: Word error rates (WER) of two masking strategies. (1) None: no speech tokens are masked. (2) Padded Audio Tokens: we mask padded speech tokens.

As shown in Table 3, these two masking strategies achieves comparable performance, indicating that masking padded speech tokens or not has little impart to model performance. We adopt strategy None, due to its easier to implement.

# 4. Choice of LLM

The choice between an instruction-tuned LLM and a base LLM (not instruction-tuned) was studies in the context of VLMs (Karamcheti et al., 2024). The conclusion is that using an instruction-tuned LLM yields no statistically significant VLM performance improvement. However, whether this holds true for SLMs has yet to be determined.

| LLM | WER ↓ |
|---|---|
| Llama3 8B Instruct | 4.89% |
| Llama3 8B | 6.18% |

Table 4: Word error rates (WER) of two variants of our model. (1) Llama3 8B Instruct: we adopt an instruction-tuned LLM. (2) Llama3 8B: we adopt a base LLM.

In Table 7, we compare the WER of two variants of our model which adopts an instruction-tuned LLM, *i.e.*, Llama3 8B Instruct and a base LLM, *i.e.*, Llama3 8B, respectively. We see that an instruction-tuned LLM achieves 1.29% lower WER. Hence, we draw different conclusion from (Karamcheti et al., 2024) in the context of SLM and proceed with Llama3 8B Instruct.

5. Choice of Training Data

| Dataset(s) | # Samples | WER ↓ |
|---|---|---|
| LibriSpeech 360 | 100K | 4.89% |
| CommonVoice11 EN | 100K | 16.70% |
| CommonVoice11 ML | 280K | 17.30% |
| LibriSpeech 360 + CommonVoice11 ML | 380K | 5.10% |
| LibriSpeech 960 + CommonVoice11 ML | 580K | 4.56% |

Table 5: Comparison between different datasets (More details of these datasets are presented in Section **TBD**). We show the number of samples from each dataset and word error rate (WER) of models trained on theses datasets.

We compare several choices of alignment training data. Specifically, we would like to see whether the model performances differently when LibriSpeech dataset and CommonVoice11 dataset are used, respectively. We also explore the impact of adding multi-lingual ASR data to our models' English ASR capability. Specifically, we examine five different mixture of training data.

1. LibriSpeech 360: 100K English ASR samples from LibriSpeech train-clean 360 hour subset.

2. CommonVoice11 EN: 100K English ASR samples from CommonVoice11 dataset.

3. CommonVoice11 ML: 100K English ASR samples and 180K German, Russian, Chinese, Spanish, French and Swahili ASR samples (30K per language) from CommonVoice11 dataset.

4. LibriSpeech 360 + CommonVoice11 ML: a combination of LibriSpeech 360 and CommonVoice11 ML

5. LibriSpeech 960 + CommonVoice11 ML: a combination of LibriSpeech 960 and CommonVoice11 ML. LibriSpeech 960 is a combination of all three subsets of LibriSpeech train-clean dataset.

As shown in Table 5, models trained on CommonVoice11 EN and CommonVoice11 ML have more than 10% higher WER than other models. We carefully examine models' predictions and the training data. It turns out that as transcripts of more than 95% of samples from these two data mixture have less than 40 tokens. The trained models suffer to handle speech clips that have more than 40 tokens (roughly 10-second long). We can also see that adding multi-lingual ASR data only slightly harms English ASR capability. As this enables our model to handle non-English clips, we move forward with data mixture that have multi-lingual ASR data.

## 3.2 INSTRUCTION TUNING

After exploring key design choices for alignment training, we proceed to conduct instruction tuning experiments.

### 3.2.1 SYNTHETIC SPOKEN INSTRUCTION DATA GENERATION

We start by playing with existing SLMs (SALMONN (Tang et al., 2023) and Qwen Audio Chat (Chu et al., 2023), to be specific). To our surprise, we existing SLMs struggles to respond to spoken instruction or answer spoken questions, even simple ones like "`can I go to the moon`?". Examples conversations are shown in Figure 4. As can be found in (Tang et al., 2023; Chu et al., 2023), while these models are trained with instruction data that contains thousands of hours of speech clips, speech clips serve as *contexts* instead of *instructions*. Instructions are provided using *text*. We hypothesize that this is the reason why they lack spoken instruction following capability.

To validate our hypothesis, we build an spoken instruction following dataset which contains 50K samples with *spoken* instructions. We create this dataset by converting 50K text

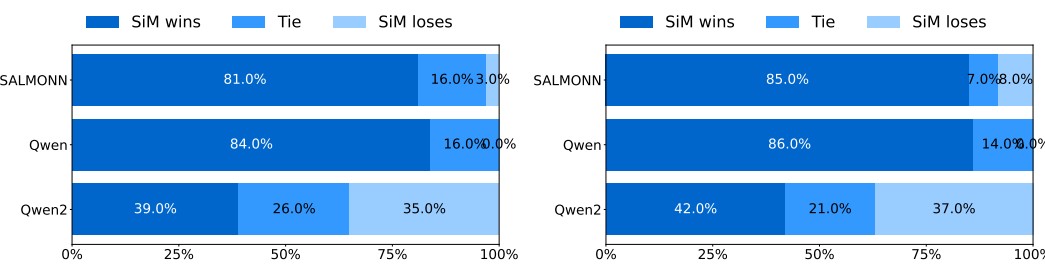

(a) Human preference evaluation, comparing SiM to 3 different existing models, *i.e.*, SALMONN, Qwen Audio Chat (Qwen) and Qwen2 Audio (Qwen2), across 100 test prompts from OpenHermes Audio dataset.

(b) Human preference evaluation, comparing SiM to 3 different existing models, *i.e.*, SALMONN, Qwen Audio Chat (Qwen) and Qwen2 Audio (Qwen2), across 100 test prompts from Alpaca Audio dataset.

Figure 2: Human preference evaluation on OpenHermes Audio dataset (left) and Alpaca Audio dataset (right).

instructions to speech with Amazon Polly Text to Speech service. We only keep convert instructions whose lengths are within 6 to 60 words and are *readable*. Text responses are generated by prompting one of the top-performing private LLMs with the original text prompts.

### 3.2.2 Comparison with Existing Models

We train our model, SiM, using a combination of (1) our spoken instruction following data with 50K samples and (2) ASR data composed of 20K samples from LibriSpeech dataset and 46K samples from CommonVoice11 dataset which include 10K English ASR samples and 36K multi-lingual ASR samples.

# 1. Spoken Instruction Following Capability

| Dataset | Model | | | | |
|---|---|---|---|---|---|
| | SALMONN | Qwen | WavLLM | SiM | Llama3 8B |
| OpenHermes ↑ | 1.05 | 1.14 | 1.07 | **2.71** ↑1.57 | 2.82 |
| Alpaca ↑ | 0.85 | 1.12 | 1.15 | **2.63** ↑1.48 | 3.37 |

Table 6: Comparison of spoken instruction following capabilities of SiM, SALMONN, Qwen Audio Chat (Qwen) and WavLLM on two datasets: OpenHermes Audio and Alpaca Audio. Scores of Llama3 8B are shown in gray as it takes transcripts instead of audio clips as inputs.

We compare our model's spoken instruction following capability with that of existing SLMs. As shown in Table 6, our model, SiM, significantly outperforms existing SLMs on two different datasets. SiM's scores are more than **125**% higher than those of the best existing SLM. Noticably, on OpenHermes Audio dataset, SiM's score is comparable to that of Llama3, which takes *textual* transcripts as input. With this said, SiM's score is still 0.74 lower than that of Llama3 on Alpaca Audio dataset. This means that there are still work that needs to be done to fully close the gap between instruction following capabilities of SLMs and LLMs.

As scores in Table 6 are obtained by using model as a judge, to draw conclusions more reliably, we have humans compares the performance of SiM with SALMONN, Qwen Audio Chat and Qwen2 Audio. We present annotators with a speech clip and textual responses generated by two different models. We ask the annotator to choose the better answer or select "`tie`" if both answers are of similar quality. Figure 2b and 2a presents human annotation results. We see similar trends as shown in Table 6.

We also compare the spoken question answering capability between SiM and existing SLMs on LLaMA Questions dataset (Nachmani et al., 2023). We see that SiM outperforms existing SLMs by a large margin. This further demonstrates the superior spoken instruction following

| Models | Accuracy ↑ |
|---|---|
| Spectron | 22.9% |
| SpeechGPT | 21.9% |
| SALMONN | 40.7% |
| Qwen | 18.0% |
| SiM | 57.7 % |

Table 7: Question answering performance of SiM, SALMONN, Qwen Audio Chat (Qwen), Spectron and SpeechT5 on LLaMA Questions dataset. Following (Nachmani et al., 2023), we adopt accuracy as our evaluation metric.

---

**LibriSpeech Dataset ASR Examples**

**[Example #1]**
**GT Transcript**: he hoped there would be stew for dinner turnips and carrots and bruised potatoes and fat mutton pieces to be ladled out in thick peppered flour fattened sauce
**SiM Prediction**: he hoped there would be stew for dinner turnips and carrots and bruised potatoes and fat mutton pieces to be ladled out in thick peppered flour fattened sauce

**[Example #2]**
**GT Transcript**: the music came nearer and he recalled the words the words of shelley's fragment upon the moon wandering companionless pale for weariness
**SiM Prediction**: the music came nearer and he recalled the words the words of shelley's fragment upon the moon wandering companionless pale for weariness

---

Figure 3: ASR examples from LibriSpeech dataset (test-clean subset).

capability of our model, thus validating our hypothesis that lacking instruction tuning data with spoken instructions is the reason for unsatisfactory spoken instruction following ability of existing SLMs.

# 2. ASR Capability

| Dataset | Model | | | | |
|---|---|---|---|---|---|
| | SALMONN | Qwen | WavLLM | SiM | Whisper |
| Librispeech test-clean ↓ | 2.4 | 1.8 | 2.0 | **3.1** ↓1.3 | 2.1 |

Table 8: ASR performance, which is measured by word error rate (WER), of SiM, SALMONN, Qwen Audio Chat (Qwen) and WavLLM on LibriSpeech dataset (test-clean subset). WER of Whisper is shown in gray as Whisper is mainly an ASR model which is not able to follow speech instruction.

We compare the ASR capability of SiM with that of existing SLMs. As shown in Table 8, Qwen achieves the lowest WER, SiM's WER is comparable to existing SLMs. Note that SiM only uses less than one thirds of ASR samples examples compared to Qwen. We also present qualitative ASR examples in Figure 3.

## 4 CONCLUSION

In this work, we conduct a thorough exploration of the design space of SLMs through rigorous experiments under comparable settings. Our key insights: (1) Surprisingly, a simple 2-layer MLP adaptor works better than more complex architectures. (2) Using an instruction-tuned LLM is favorable over a base LLM. (3) A combination of the Common-Voice and LibriSpeech datasets is a suitable choice for alignment training. In addition, we uncover that including *spoken* instructions in the instruction tuning data is essential to enable SLMs to follow spoken instructions effectively, a capability lacking in existing models.

Leveraging these insights and our synthetic spoken instruction data, we present SiM, our SLM model. SiM significantly outperforms existing SLMs in spoken instruction following

while achieving comparable performance in ASR. Our work provides an effective recipe for developing high-performing SLMs capable of understanding and following spoken instructions.

---

**Alpaca Audio Dataset Examples**

---

**[Example #1]**
`Text Instruction`: Please obey the instructions that were provided in the audio.
`Audio Instruction`: Write an explanation of why wearing a helmet is important.

---

`SiM`: Wearing a helmet is important because it protects the head from injury in the event of a crash or fall. Helmets are designed to absorb the impact of a collision and distribute the force of the impact across the head, reducing the risk of serious injury or death. They are especially important for motorcyclists, cyclists, and skiers, as they are more vulnerable to head injuries than drivers of cars or trucks. Additionally, wearing a helmet can also help prevent concussions and other head injuries that can have long-term effects on the brain. Overall, wearing a helmet is a simple and effective way to protect oneself from serious head injuries.

`SALMONN`: Write an explanation of why wearing a helmet is important.

`Qwen Audio Chat`: The lyrics are "Write an explanation of why wearing a helmet is important to me"

`Qwen2 Audio`: Wearing a helmet is important for several reasons:

1. Protection: Helmets provide a layer of protection between your head and any hard objects that may be thrown or fall towards you while participating in activities such as cycling, skateboarding, or motocross.

2. Safety: Helmets can help prevent serious injuries to the head, including concussions, skull fractures, and brain trauma. These injuries can be life-changing or even fatal, so wearing a helmet can greatly reduce your risk of sustaining such injuries.

3. Compliance: In many sports and activities, wearing a helmet is mandatory. Failure to wear a helmet can result in fines, penalties, or other sanctions from organizations or governing bodies.

4. Style: Helmets come in a variety of styles and designs, so you can choose one that matches your personal taste or the sport or activity you're participating in.

Overall, wearing a helmet is an essential piece of protective gear that can save your life or significantly reduce your risk of injury.

---

**[Example #2]**
`Text Instruction`: Please make sure to follow the instructions provided in the audio.
`Audio Instruction`: Are Seattle and Vancouver closer to one another than Boston and New York?

---

`Sim`: Yes, Seattle and Vancouver are closer to each other than Boston and New York. Seattle is located in the state of Washington in the Pacific Northwest region of the United States, while Vancouver is located in the province of British Columbia in Canada. Boston is located in the state of Massachusetts in the Northeastern United States, while New York is located in the state of New York in the Northeastern United States. The distance between Seattle and Vancouver is approximately 140 miles (225 km), while the distance between Boston and New York is approximately 215 miles (346 km).

`SALMONN`: Are seattle and vancouver closer to one another than boston and new york?

`Qwen Audio Chat`: The answer is no.

`Qwen2 Audio`: The distance between Seattle and Vancouver is approximately 270 miles (435 kilometers), while the distance between Boston and New York is about 268 miles (419 kilometers). Therefore, Seattle and Vancouver are closer to one another than either of these pairs.

---

**[Example #3]**
`Text Instruction`: Please make sure to follow the instructions provided in the audio.
`Audio Instruction`: Classify the following movie genres: comedy, action, drama.

`Sim`: I can provide you with a general classification of movies based on their genre, comedy, action, and drama.

- Comedy: movies that are intended to be humorous and entertaining, often featuring comedic characters and situations. Examples include "the hangover," "the 40-year-old virgin," and "bridesmaids."

- Action: movies that feature high levels of violence, suspense, and excitement, often with a focus on action sequences and stunts. Examples include "die hard," "the matrix," and "the dark knight."

- Drama: movies that focus on the emotional and psychological struggles of the characters, often featuring complex storylines and character development. Examples include "the social network," "12 angry men," and "the departed."

`SALMONN`: Classify the following movie genres: comedy, action, drama.

`Qwen Audio Chat`: The lyrics are "Classify the following movie genres Comedy, Action, Drama".

`Qwen2 Audio`: The movie genres are comedy, action, and drama.

---

Figure 4: Model outputs given test prompts from Alpaca Audio dataset. We compare outputs of SiM, SALMONN, Qwen Audio Chat and Qwen2 Audio.

---

**OpenHermes Audio Dataset Examples**

---

**[Example #1]**
**Text Instruction:** Please make it a point to follow the instructions from the audio.
**Audio Instruction:** Suggest two strategies for building trust with a client.

---

**SiM:** 
1. Be transparent and open with the client: Building trust with a client requires being transparent and open with them. This means being honest and upfront about your services, pricing, and any potential issues that may arise. It also means being responsive to their questions and concerns, and keeping them informed throughout the process.

2. Follow through on commitments: One of the most important ways to build trust with a client is to follow through on your commitments. This means delivering high-quality work on time, and meeting any deadlines that you have agreed upon. It also means being reliable and dependable, and following through on any promises you have made.

**SALMONN:** Suggests two strategies for building trust with a client.

**Qwen Audio Chat:** The speaker says in English: "Suggests two strategies for building trust with a client".

**Qwen2 Audio:** 战略一：与客户建立长期的良好关系，保持沟通渠道畅通。战略二：深入了解客户需求，提供个性化服务，让客户感受到贴心的关怀。

---

**[Example #2]**
**Text Instruction:** Please comply with the instructions given in the audio clip.
**Audio Instruction:** Calculate the mean of the following numbers: 11, 15, 13, 17 and 19.

---

**Sim:** The mean of the following numbers is:
$11 + 15 + 13 + 17 + 19 = 75$
$75/5 = 15$

**SALMONN:** Calculate the mean of the following numbers: eleven, fifteen, thirteen, seventeen, and nineteen.

**Qwen Audio Chat:** The answer is no.

**Qwen2 Audio:** The mean of the numbers 11, 15, 13, 17, and 19 is:
$(11 + 15 + 13 + 17 + 19)/5 = 80/5 = 16$
Therefore, the mean of these numbers is 16.

---

**[Example #3]**
**Text Instruction:** Please make sure to follow the instructions provided in the audio.
**Audio Instruction:** Create a function in Python which takes a list and finds the second highest number in it."

---

**Sim:**

```python
1  def second_highest(lst):
2      lst = list(set(lst))
3      lst.sort()
4      if len(lst) < 2:
5          return None
6      else:
7          return lst[-2]
```

**SALMONN:** Classify the following movie genres: comedy, action, drama.

**Qwen Audio Chat:** The lyrics are "Classify the following movie genres Comedy, Action, Drama".

**Qwen2 Audio:** Here's a Python function that takes a list as input and returns the second highest number in it:

```python
1  python
2  def find_second_highest_number(numbers):
3      if len(numbers) < 2:
4          return None
5
6      first_highest = max(numbers)
7      second_highest = min(numbers)
8
9      return second_highest
```

The function first checks if the length of the list is less than 2, in which case it returns None since there cannot be a second highest number.
If the length of the list is greater than or equal to 2, the function finds the maximum number $first_highest$ and the minimum number $'second_highest'$ in the list using the built-in 'max' and 'min' functions respectively.
Finally, the function returns the second highest number found.
Here's an example usage of the function:

```python
1  python
2  numbers = [5, 9, 3, 7, 2, 8]
3  print(find_second_highest_number(numbers)) # Output: 8
```

Note that ties for the second highest number

---

Figure 5: Model outputs given test prompts from OpenHermes Audio dataset. We compare outputs of SiM, SALMONN, Qwen Audio Chat and Qwen2 Audio.

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
