# OpenReview forum: "An Exploration of Speech Conditioned Large Language Models (SLMs)"
_ICLR.cc/2025/Conference — Submitted to ICLR 2025_

### Official Review · Reviewer_QRHM · 2024-10-29

**Soundness:** 2
**Presentation:** 2
**Contribution:** 2
**Rating:** 3
**Confidence:** 4

**Summary:**

This paper explores among some of the architectural choices and training recipes of the speech-conditioned language models(SLM). Specifically, the paper compares the architectural choices of the adaptor in controlled experiments and comes up with an optimal alignment training recipe including data/backbone LLM selection/encoder masking strategy. Following that, the author proposes to apply a new spoken instruction tuning dataset in training to obtain better spoken instruction tuning performance.

**Strengths:**

- Part of this paper aims to conduct a systematic study on the design choices of SLMs under a comparable set-ups, which could potentially serve as great reference for the community in the future work.

- This paper finds that there are opportunities to improve the spoken instruction following capability in existing open-source SLMs and propose to address that by adding in-domain data.

**Weaknesses:**

**Lack of Novelty**

The author proposes to mixing the synthetic spoken instruction following dataset in the training to improve the spoken instruction following performance. The resulting model, SiM, outperforms other counterparts in OpenHermes and Alpaca benchmark. While this indicates advancement in speech interaction of the chatbot application, the improvement seems to solely come from adding in-domain data instead of any novel methodology, which brings questions to the novelty of the proposed methods. To prove that other SLMs are indeed performing worse in that task, various forms text prompts should be examined and the author may report the best scores among them.

**Limitation**

The paper aims to explore the design space of SLM. However, the conclusion from this work is not generic enough. For example:

+ The architecture is limited within the form of encoder-adaptor-LM, where the encoder embeds the speech input in the continuous space. The findings in this paper couldn't be applied to the SLMs based on discrete speech units.
+ The target of "Alignment Training" is limited to ASR task. Whether the design choice selected by lowest WERs also leads to better performance is not examined in this paper.

**Lack of details**

I would suggest disclosing more training details for reproducibility including common hyper parameters, training infra, etc. Also, for the synthetic dataset, since it's not open-sourced, examples should be given.

**Questions:**

- Paper requires more proof-reading, for example: Appendix is missing (mentioned in Section 3.1), also, the caption of Table 5 says *"More details of these datasets are presented in Section TBD"*.
- For consistent presentation, the conclusion can be made for each subsection in Section 3.1 (like the conclusion#1 in 3.1-#1)

---

### Official Review · Reviewer_xo4R · 2024-11-03

**Soundness:** 2
**Presentation:** 2
**Contribution:** 1
**Rating:** 3
**Confidence:** 4

**Summary:**

The paper explores the speech-LLM model design architectures. Specifically, the exploration is mainly in SALMONN or Qwen Audio like designs but does not cover other design options to introduce speech to LLM. The paper further introduces a form of speech instruction tuning and shows some competitive results.

**Strengths:**

Given the SALMONN or Qwen Audio like designs, the paper seems to be thorough in exploring fine-grain model architecture choices.

**Weaknesses:**

1. The paper claims to thoroughly explore SLM design space, but in fact only explore model architecture choices under the category of SALMONN or Qwen Audio like Speech-LLM, whose scope is very limited and may not be able to extend to other high level architecture designs. Besides the architecture used by the paper, there are more different Speech-LLM models like AudioPaLM, Moshi, AudioFlamingo, BESTOW, etc. The paper fails to analyze and compare the architecture differences and motivate the reason to use the proposed architecture.
2. Almost all of the components in the paper are not new, which limits its unique contributions: i) the architecture itself follows SALMONN or Qwen Audio ii) the study in adapter architecture and trainable module is very similar to previous works like https://ieeexplore.ieee.org/stamp/stamp.jsp?arnumber=10445874 iii) the way to build the synthetic spoken instruction following dataset is not new, e.g. https://arxiv.org/pdf/2310.13289 and https://arxiv.org/abs/2309.09843 and https://arxiv.org/abs/2406.12946
3. The comparison of different design choices is completely based on ASR. Whether these conclusions can hold on to other tasks is a big unsolved problem, which limits the value of these conclusions, considering one of the central value of Speech-LLM is its speech instruction following and speech understanding ability . Moreover, the authors should also consider translation tasks.
4. The way to evaluate speech instruction following ability is limited. The authors should consider public benchmarks like AIR-bench, Dynamic Superb or SD-Eval.
5. Speech instruction SLM and text instruction SLM are two different downstream applications or problems. Most of the baselines the paper tries to compare with is designed for text instruction SLM, which is unfair to compare on speech instruction testsets, especially considering the proposed work is mainly designed for speech instruction.

**Questions:**

Please resolve the concerns in Weakness.

---

### Official Review · Reviewer_5ppY · 2024-11-03

**Soundness:** 2
**Presentation:** 3
**Contribution:** 2
**Rating:** 3
**Confidence:** 4

**Summary:**

This paper investigates the design space of Speech-Conditioned Large Language Models (SLMs) with the goal of enhancing their ability to comprehend and follow spoken instructions. The study observes that current SLMs often struggle with spoken instruction adherence and addresses this gap by creating a synthetic dataset for spoken instruction following. Key design choices influencing model performance are identified, and an effective training approach is proposed, yielding SLMs with improved ASR capabilities and enhanced spoken instruction-following proficiency.

**Strengths:**

1. The paper makes a significant contribution by constructing a spoken instruction-following dataset with 50K samples. Utilizing this dataset, the proposed SiM model demonstrates impressive spoken instruction-following capabilities, achieving substantial performance gains over existing SLMs.
2. Through thorough experimentation, the paper explores the design space of SLMs, examining critical factors like adaptor architecture and trainable modules. These findings offer valuable insights and serve as a practical reference for the academic community in the design and training of SLMs.

**Weaknesses:**

1. While the paper aims to provide an effective framework for developing SLMs by thoroughly exploring the design space, its focus on only ASR and instruction-following tasks significantly narrows the research scope, limiting the general applicability of its findings.
2. The ASR component lacks novelty, as it has been extensively studied in the field. Prior work, such as "An Embarrassingly Simple Approach for LLM with Strong ASR Capacity" (Ma et al., 2024), has already benchmarked various combinations of large language models and speech encoders, establishing optimized LLM-based ASR systems. This limits the originality of the ASR contributions in this paper.
3. In investigating the Adaptor Architecture, the study does not control the audio token reduction rate (ATR), which varies across experiments. With an ATR greater than 1, downsampling audio tokens likely leads to information loss and potential performance degradation. This lack of control limits the validity of conclusions drawn from these experiments.
4. The experimental comparisons are somewhat limited, as they only involve Llama3 8B Instruct and Llama3 8B with a simple masking strategy and solely rely on the Whisper encoder. This narrow scope restricts the conclusions, as a broader range of comparisons with different models and encoders would yield more generalizable insights.
Furthermore, the title of this paper seems inconsistent in the pdf and the system: "An Exploration of Speech Conditioned Large Language Models (SLMs)" vs "An Exploration of the Design Space of Speech-Conditioned Large Language Models (SLM)"

**Questions:**

1. Regarding the choice of training data, could the authors clarify the rationale for evaluating all models on the LibriSpeech dataset (test-clean subset), given that this introduces a domain mismatch when compared with Common Voice data? Evaluating on a more consistent dataset may yield fairer results.
2. The paper does not compare the proposed SLM model's ASR performance against other existing SLM models. Could the authors provide these comparisons to better demonstrate the effectiveness of the proposed approach in the ASR task?
3. The inclusion of multilingual ASR samples for training SiM is somewhat unclear, as the benchmarking datasets mentioned, such as OpenHermes, Alpaca, LLaMA Questions, and Librispeech, are all English-based. Could the authors elaborate on the motivation and intended impact of multilingual data in this context?
4. While the paper notes that "SiM uses less than one-third of the ASR samples compared to Qwen," the model's WER is significantly higher than Whisper's by an absolute margin of 1. Could the authors discuss the trade-offs in sample efficiency versus performance and address whether additional tuning might close this performance gap?

---

### Official Review · Reviewer_sfZa · 2024-11-04

**Soundness:** 2
**Presentation:** 2
**Contribution:** 1
**Rating:** 3
**Confidence:** 4

**Summary:**

This paper presents a comprehensive exploration of the design space for spoken language models (SLMs) and offers a practical guide for their training. The authors develop SiM, a model that demonstrates strong performance in both automatic speech recognition (ASR) and spoken instruction following tasks.

**Strengths:**

This paper introduces the synthetic spoken instruction following dataset to enhance the ability of SLMs to follow speech-based instructions.

**Weaknesses:**

1. In Section 3.1 (Alignment Training), the experiments on model architecture and training strategy choices are evaluated solely on the LibriSpeech test-clean. Since LibriSpeech test-clean contains only clean audiobook data, the experiments seem more like an exploration of the best-fitting strategy for this specific domain. This issue is particularly evident in Section 3.1.5 (Choice of Training Data), where the authors attempt to explore data-mixing strategies between CommonVoice and LibriSpeech but rely only on evaluation results from LibriSpeech test-clean. Given the distinct domains of Common Voice and LibriSpeech, this approach may lead to significant biases in the final conclusions. I recommend adding more diverse test data, such as LibriSpeech test-other and CommonVoice, to enhance the robustness and reliability of the experimental conclusions in this section.
2. The final SiM model architecture largely aligns with mainstream MLLM architectures like LLaVA[1] and SLAM-ASR[2]. In Section 3.2.1, the authors find that using Synthetic Spoken Instruction Data (SiM) improves spoken instruction following compared to models such as Qwen2-Audio and SALMONN, which were not trained with spoken instruction data. However, this result seems self-evident. The use of synthetic spoken instruction data to enhance instruction-following ability has already been proposed in prior work, such as SpeechGPT[3], though SpeechGPT used Chain-of-Modality for this purpose, while SiM does not generate corresponding speech transcriptions. Overall, the paper’s main methods lack novelty and offer few technical contributions.

[1] Liu, H., Li, C., Wu, Q., & Lee, Y. J. (2024). Visual instruction tuning. Advances in neural information processing systems, 36.

[2] Ma, Z., Yang, G., Yang, Y., Gao, Z., Wang, J., Du, Z., ... & Chen, X. (2024). An Embarrassingly Simple Approach for LLM with Strong ASR Capacity. arXiv preprint arXiv:2402.08846.

[3] Zhang, D., Li, S., Zhang, X., Zhan, J., Wang, P., Zhou, Y., & Qiu, X. (2023). Speechgpt: Empowering large language models with intrinsic cross-modal conversational abilities. arXiv preprint arXiv:2305.11000.

**Questions:**

1. What is the source of the textual data used to synthesize the spoken instruction following dataset? It appears that this information is not provided in the paper.
2. It is confusing that training Adaptor+Whisper results in a much higher WER than either Adaptor+LLM or Adaptor alone. Could the authors provide some explanation for this outcome?

---

### Meta-Review · Area_Chair_oJCq · 2024-12-16

**Metareview:**

This paper provides an empirical study of the architecture design of Speech-Conditioned Large Language Models (SLMs), showing simple MLP adaptor works even better and instruction-tuned LLM is better than a base LLM. Furthermore, the authors argued some existing SLMs cannot work well for spoken query as the models were trained with text instructions. To resolve this issue, the authors synthesized 50k samples spoken instructions by converting the text instruction into speech, which later is shown to be effective to boost the spoken instructions following capabilities.

While the work is of practical benefit to the field, overall it lacks of novelty and has limitations.
•	The exploration of model architecture is constrained to the SLMs which use an adaptor to connect the speech encoder with LLMs, such as SALMONN and Qwen Audio. In the field, there is another type of SLMs such as AudioPaLM which take the discrete audio token as input. Therefore, the architecture exploration is not generalized to all SLMs.
•	Furthermore, there are already similar exploration works for the model architecture as called out by Reviewer 5ppY and Reviewer xo4R.
•	The work is mainly limited to exploring the impact to ASR, while other speech tasks should be also examined. It raises the question again whether the conclusions can be generalized.
•	Even the building of spoken instruction data is not new, as called out by Reviewer xo4R.

**Additional Comments On Reviewer Discussion:**

There is no author/reviewer discussion.

---

### Decision · Program_Chairs · 2025-01-22

Reject